# Transfer Hydrogenation of Vinyl Arenes and Aryl Acetylenes with Ammonia Borane Catalyzed by Schiff Base Cobalt(II) Complexes

**DOI:** 10.3390/ijms25084363

**Published:** 2024-04-15

**Authors:** Maciej Skrodzki, Maciej Zaranek, Giuseppe Consiglio, Piotr Pawluć

**Affiliations:** 1Faculty of Chemistry, Adam Mickiewicz University, Uniwersytetu Poznańskiego 8, 61-614 Poznań, Poland; maciej.skrodzki@amu.edu.pl; 2Centre for Advanced Technologies, Adam Mickiewicz University, Uniwersytetu Poznańskiego 10, 61-614 Poznań, Poland; m.zaranek@amu.edu.pl; 3Department of Chemical Science, University of Catania, Via S. Sofia 64, 95125 Catania, Italy

**Keywords:** transfer hydrogenation, cobalt catalysis, ammonia borane, alkene, alkyne, Schiff-base ligand

## Abstract

A series of bench-stable Co(II) complexes containing hydrazone Schiff base ligands were evaluated in terms of their activity and selectivity in carbon-carbon multiple bond transfer hydrogenation. These cobalt complexes, especially a Co(II) precatalyst bearing pyridine-2-yl-N(Me)N=C-(1-methyl)imidazole-2-yl ligand, activated by LiHBEt_3_, were successfully used in the transfer hydrogenation of substituted styrenes and phenylacetylenes with ammonia borane as a hydrogen source. Key advantages of the reported catalytic system include mild reaction conditions, high selectivity and tolerance to functional groups of substrates.

## 1. Introduction

The use of first-row transition metal compounds as homogeneous catalysts has been developed for more than 20 years, leading to success in the field of synthetic organic chemistry by allowing many transformations [1,2,3]. The higher abundance of these metals compared to platinum group ones offers opportunities to develop cost-effective catalysts as well as to discover unique reactivity resulting from their different stereoelectronic properties. It is a combination of these factors, along with lower prices and generally lower environmental concerns, that make 3d electron metals and their complexes important subjects of research, giving rise to an ever-increasing number of reports on their catalytic properties. A considerable number of these complexes contain 3N donor ligands, whose use often contributes to the overall efficiency of such catalytic systems. However, it is worth stressing that the synthesis of such ligands can be cumbersome and expensive, and the need for accessibility and sustainability calls for new catalysts to have a coordinating environment as simple and rational as possible while maintaining high efficacy and selectivity.

In recent years, much attention has been paid to the use of cobalt catalysts for alkene and alkyne hydrogenation. However, cobalt catalysis faces challenges due to the low crystal field stabilization energy, the possibility of varying spin states, a propensity for 1-electron chemistry, and the unpredictable catalytic performance of cobalt ions, similar to other first-row transition metals. This contrasts with the more predictable nature of catalysis involving their 4d or 5d congeners. Thus, the design of ligands is crucial for modulating catalytic activity and selectivity, as they adjust the cobalt coordination environment. Specifically, pincer ligands, including NNN, NNP, and PNP types, have been frequently employed for their planar coordination mode. Notably, Budzelaar and colleagues reported a bis(imino)pyridine cobalt complex for hydrogenating 1- and 2-alkenes [4]. Similarly, Chirik’s team demonstrated the effectiveness of chiral bis(imino)pyridine cobalt complexes in the enantioselective hydrogenation of gem-disubstituted olefins [5]. Hanson’s group introduced a bis(phosphino)amino cobalt complex for the efficient hydrogenation of alkenes and carbonyl compounds [6], while Lu and co-workers improved the asymmetric hydrogenation of 1,1-diarylethenes using oxazoline iminopyridine-cobalt complexes [7]. Further research has acknowledged the suitability of various bidentate and tetradentate ligands for enhancing the activity of cobalt complexes in olefin hydrogenation [8,9,10].

Transfer hydrogenation utilizing ammonia borane has emerged as a viable alternative to conventional hydrogenation, attributed to its high efficiency, excellent atom economy, non-toxicity, and environmental friendliness [11]. Encouraged by successful instances of simple alkene transfer hydrogenation using Co(II) complexes with nitrogen ligands and ammonia borane as the hydrogen donor [12], we were motivated to examine the potential of our tridentate Schiff-base cobalt(II) complexes in transfer hydrogenation reactions.

## 2. Results and Discussion

Taking into account the exceptionally good results of using Schiff base cobalt complexes in hydrosilylation that we previously reported [13,14,15], these compounds were chosen for examination in the catalytic hydrogenation of alkenes and alkynes. Schiff base ligands were synthesized in a well-known two-step procedure of, first, substitution of heteroaromatic chlorides (depicted in blue) with methylhydrazine, followed by condensation of the obtained hydrazines with heteroaromatic aldehydes (purple) that produce the desired ligands, usually with excellent yields (scheme) [13,14,15]. These ligands were then subjected to simple complexation with cobalt(II) chloride.



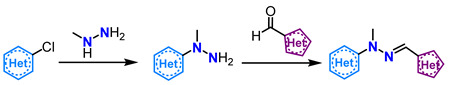



The compositions of 16 ligands and the preliminary results of the catalytic activity of their corresponding Co(II) complexes in the transfer hydrogenation of α-methylstyrene are shown in Table 1. To simplify the reaction system, the ammonia borane adduct was chosen as the hydrogen source. This reagent is easily accessible and relatively non-toxic while offering a good balance between shelf stability and reactivity [16]. The catalytic system consisted of 1 mol% of a precatalyst activated by adding 3 mol% of lithium triethylborohydride.

Based on the results shown in Table 1, it is difficult to draw unambiguous conclusions about the relationship between the structures of the ligands and the catalytic activity of the corresponding cobalt complexes. The only visible trend is that the presence of the thiophen-2-yl substituent (L1, L5, L9, L13) turned out to be less beneficial than any of the imidazolyl configurations. Nonetheless, L8 has been chosen for further investigation as not only did its use result in the highest conversion (93%), but it also happens to contain the simplest (and arguably the cheapest) hydrazine-side substituents, i.e., pyridin-2-yl. It was also determined that a reaction without lithium triethylborohydride does not proceed. Previously, we have discussed the possible influence that borohydride can exert on Schiff base ligands [14]; however, it is not possible to ascertain the role of LiHBEt_3_ in the case of the catalytic system described here.

Having chosen the catalyst, we decided to try different hydrogen sources. To account for the necessary headroom, the amount of catalyst for these trials was reduced to 1 mol%. As the summary provided in Table 2 shows, no other potential hydrogen source from among ethanol, methanol, hydrogen, sodium tetrahydroborane, or phenylsilane proved anywhere near as effective as the ammonia borane adduct used initially. Not entirely surprisingly, phenylsilane gave only products of hydrosilylation instead of hydrogenation. The ineffectiveness of alcohols under the conditions of our process is most likely due to their reactivity toward the borohydride activator, which does not allow for the activation of the precatalyst. Furthermore, an attempt to decrease the temperature resulted in a significant drop in alkene conversion.

As stated in previous work [13], the most significant impact on reaction efficiency is mainly driven by the imidazole/thiophene fragment of the ligand. The role of borohydride is unclear; however, its presence is necessary for the completion of the reaction. Based on our previous results obtained for different Schiff base cobalt complexes, we have concluded that borohydride might be involved in the modification of the ligand structure [14]. Lower conversions were observed with application complexes with 2-thiophene moieties (50% to 88% yield). The application of complexes with 4-imidazole or 2-imidazole (N-H bonds) resulted in better yields; however, the highest yield of isopropylbenzene (cumene) was observed for reaction with the pyridine-methylhydrazine-(4-Me)imidazole Co(II) precatalyst. Our group has previously found this precatalyst to be highly active in the hydrosilylation of aromatic olefins [15].

After the initial trials with the precatalyst, optimization with the pyridine-methylhydrazine-(4-Me)imidazole Co(II) precatalyst was conducted (Table 2). For this purpose, several hydrogen donors were tested in the reaction with α-methylstyrene in the presence of cobalt(II) precatalyst and LiHBEt_3_. The best results were observed for 2% of Co(II) precatalyst, 8% of LiHBEt_3_ and ammonia borane as the hydrogen donor. A further increase in catalyst loading was not necessary, as the reaction has already yielded 99%. Lowering the temperature significantly decreased the conversion of alkene. No conversion of alkene was observed in reactions with simple alcohols as hydrogen donors. The alcohol consumed the loading of lithium triethylborohydride prior to its reaction with the precatalyst, halting the proper activation. Additionally, no reaction occurred in the presence of hydrogen or sodium tetrahydroborate. The reaction with phenylsilane yielded only alkene hydrosilylation products with a low conversion of alkene.

Having obtained the optimized conditions for α-methylstyrene hydrogenation, various unsaturated compounds were then tested. The results are summarized in Table 3 and exemplary chromatograms can be found in the Appendix A. The best results were observed for simple styrene-bearing methyl substituents on the phenyl group. A total conversion of 4-methylstyrene was confirmed; however, 1-methoxy-4-(prop-1-en-1-yl)benzene, the internal alkene, was unable to react under the given conditions. A wide scope of halogen substituents was evaluated. 4-trifluoromethylstyrene reacted with total conversion and 4-fluorostyrene with 86% conversion, but to our surprise, the methyl substituent at the α-position in fluorostyrene stopped the reaction. A similar reactivity was observed for 4-bromostyrene and 4-bromo-α-methylstyrene. 4-bromostyrene reacted in a moderate manner, whereas the α-methyl analogue did not react at all. 4-Chlorostyrene reacted with a 57% conversion. No conversions were observed for (4-chloromethyl)styrene and 4-aminostyrene. No difference was seen for the application of (E) or (Z)-stilbene; in both reactions, the conversion was equal to 21%. Hydrogenation involving allylbenzene resulted in 49% of substrate conversion. The selected aromatic alkynes were tested under similar conditions. 4-tertbutylphenylacetylene and 4-ethynylanisole reacted with poor yield, affording semi-hydrogenation products with 13% and 17% yield, respectively. Greater reactivity was observed for 1,2-diphenylacetylene–63%. The experiments, with a duration of 4 days, involving 4-bromophenylacetylene and (4-phenyl)phenylacetylene, afforded semi-hydrogenation products with a 99% yield. No products were observed for reaction with 4-aminophenylacetylene. Despite the noticeable reactivity of alkynes, efficient stereoselective hydrogenation is still unachievable under the tested conditions.

## 3. Materials and Methods

### General Procedure for the Transfer Hydrogenation of Alkenes

In a typical reaction, alkene (0.5 mmol), catalyst (2 mol%), LiHBEt_3_ (8 mol%) NH_3_·BH_3_ (0.6 mmol) and THF (0.5 mL) were placed in a Schlenk tube under an atmosphere of argon. At this point, mesitylene was added to the mixture as an internal standard. The reaction mixture was stirred for 20 h at 60 °C. Then, 5 mL of diethyl ether was added to the reaction solution. The solids were filtered through a silica plug. The yield was determined by gas chromatography (GC), or crude product, directly loaded on a flash column chromatography to obtain a pure product whose identity was confirmed by GC-MS analysis. GC-MS analyses of selected products are included in the Appendix A.

## 4. Conclusions

Within the article, the recognition of the activity of the cobalt(II) Schiff base complex as a precatalyst for the transfer hydrogenation of substituted styrenes and phenylacetylenes has been shown. The presented method of hydrogenation features the application of the ammonia borane adduct as a convenient and easy way to synthesize a source of hydrogen. Within the examination of the reaction scope, several unsaturated compounds were tested. Despite the good activity in hydrogenation of simple aryl compounds, the reactivity of subsequent systems towards internal unsaturated bonds is still challenging.

## Figures and Tables

**Table 1 ijms-25-04363-t001:** The catalytic activity of Co(II) complexes in the hydrogenation of α-methylstyrene.

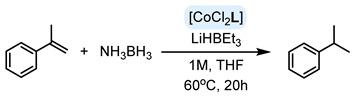
	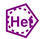	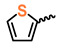	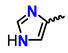	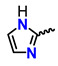	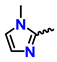
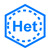	
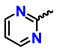	L1, 75%	L2, 92%	L3, 52%	L4, 92%
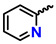	L5, 50%	L6, 91%	L7, 81%	L8, 93%
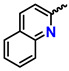	L9, 73%	L10, 75%	L11, 90%	L12, 84%
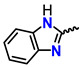	L13, 88%	L14, 82%	L15, 90%	L16, 82%

Calculated yield of isopropylbenzene (cumene) in hydrogenation of α-methylstyrene in the presence of cobalt(II) precatalysts. Conditions: Alkene 0.5 mmol, NH_3_BH_3_ 0.6mmol, 1 mol% of Co complex, 3 mol% LiHBEt_3_, 1M solution of alkene in tetrahydrofuran (THF), 60 °C, 20 h. Calculated by GC with mesitylene (0.166 mmol) as internal standard.

**Table 2 ijms-25-04363-t002:** Efficiency of different hydrogen sources in the hydrogenation of α-methylstyrene.

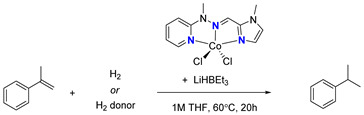			
Run	H_2_ Donor	[Co] mol%/LiHBEt_3_ mol%	GC Yield of Cumene
1	NH_3_BH_3_	1%/3%	93%
2	NH_3_BH_3_	2%/8%	99%
3	MeOH	1%/3%	0%
4	EtOH	1%/3%	0%
5	H_2_	1%/3%	0%
6	NaBH_4_	1%/3%	<1%
7	H_3_SiPh	1%/3%	0% ^a^
8	NH_3_BH_3_	1%/3%	11% ^b^

Calculated yield of cumene in hydrogenation of α-methylstyrene in the presence of cobalt(II) precatalyst. Conditions: Alkene 0.5 mmol, H_2_ donor 0.6 mmol, 1 M solution of alkene in THF, 60 °C, 20 h. Calculated by GC-MS with mesitylene (0.166 mmol) as internal standard. ^a^ Products of hydrosilylation were observed. ^b^ Reaction with α-methylstyrene at room temperature.

**Table 3 ijms-25-04363-t003:** Hydrogenation of vinyl arenes and aryl acetylenes in the presence of cobalt(II) precatalyst.

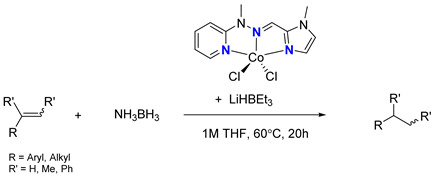
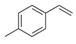	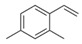	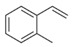	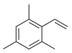
100%	100%	95%	86%
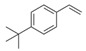	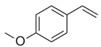	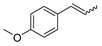	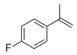
100%	100%	0%	0%
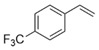	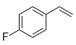	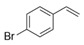	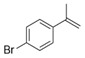
100%	86%	75%	0%
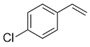	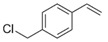	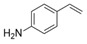	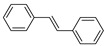
57%	0%	0%	21%
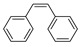	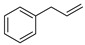	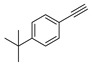	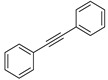
21%	49%	13%	63%
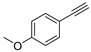	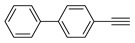	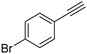	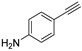
17%	100% ^a^	100% ^a^	0%

Conditions: catalyst (2 mol%), LiHBEt_3_ (8 mol%), Alkene/Alkyne 0.5 mmol, NH_3_BH_3_ 0.6 mmol, 1M solution of alkene/alkyne in THF, 60 °C, 20 h. GC-MS yields calculated with mesitylene (0.166 mmol) as internal standard. For reaction with alkynes as substrates, yield of semi-hydrogenation is given. ^a^ Reaction time–96 h.

## Data Availability

No new data were created.

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
