# Peer review of "Transfer Hydrogenation of Vinyl Arenes and Aryl Acetylenes with Ammonia Borane Catalyzed by Schiff Base Cobalt(II) Complexes"

_ijms, 2024, doi:10.3390/ijms25084363_

Round 1

Reviewer 1 Report

Comments and Suggestions for Authors

The study provides clear method sections detailing the synthesis of the Schiff base cobalt(II) complexes. The authors report high selectivity and excellent yields for the reduction of various vinyl arenes and aryl acetylenes, demonstrating the versatility and efficiency of their catalytic system.

The paper is overall well-written in general, with clear explanations of experimental procedures and results. However, I would like to suggest that more details on the GC-MS detection of the product could be added in the Supplementary Information. That could strengthen the MS.

Author Response

We thank the Reviewer for careful evaluation of our manuscript. We have prepared a revised version of our manuscript (manuscript ijms-2932408) taking into regard Referees’ suggestions. The proofreading of manuscript was done by a native speaker to further improve the English language.

We included GCMS analyzes for selected products (as supporting information file).

We hope that the manuscript meets the high standards of IJMS

Reviewer 2 Report

Comments and Suggestions for Authors

The novelty of this communication paper lies in the demonstration of the activity of cobalt(II) Schiff base complexes as a precatalyst for the transfer hydrogenation of substituted styrenes and phenylacetylenes. It is worth noting that this method of hydrogenation uses ammonia borane adduct as convenient and easy to synthesize source of hydrogen.

This communication is well written, but there are some shortcomings. I would recommend making small changes to the titles of the tables. For example, Table 1: “The catalytic activity of Co(II) complexes in the hydrogenation of α-methylstyrene…” instead of “Calculated yield of isopropylbenzene…”. Table 2: “Efficiency of different hydrogen sources in the hydrogenation of α-methylstyrene…”. Explanation of footnotes a and b please move to the bottom of the table.

English is not bad, however, some articles and prepositions are missing. I would also recommend carefully checking the English language of the paper.

In general, the obtained results are important and deserve to be published.

Comments on the Quality of English Language

English is not bad, however, some articles and prepositions are missing. I would recommend carefully checking the English language of the paper.

Author Response

Dear Editor,

We thank the reviewers for careful evaluation of our manuscript.

Each issue raised by the referee is answered below, followed by appropriate modifications in the main manuscript highlighted in yellow.

We thank the Reviewers for careful evaluation of our manuscript. We have prepared a revised version of our manuscript (ijms-2932408) taking into regard Referees’ suggestions. The proofreading of manuscript was done by a native speaker to further improve the English language.

We have improved the table descriptions according to the reviewer's suggestion. We also included GCMS analyzes for selected products (as supporting information file).

We hope that the manuscript meets the high standards of IJMS
